# Headlines win elections: Mere exposure to fictitious news media alters voting behavior

Roland Pfister[1,2,3]*, Katharina A. Schwarz[3]*, Patricia Holzmann[3], Moritz Reis[3], Kumar Yogeeswaran[4], Wilfried Kunde[3]

**1** General Psychology, Trier University, Trier, Germany, **2** Institute for Cognitive and Affective Neuroscience (ICAN), Trier University, Trier, Germany, **3** Department of Psychology III, University of Wuerzburg, Wuerzburg, Germany, **4** School of Psychology, Speech, and Hearing, University of Canterbury, Christchurch, New Zealand

* roland.pfister@uni-trier.de (RP); katharina.schwarz@uni-wuerzburg.de (KAS)

## Abstract

Repeatedly encountering a stimulus biases the observer's affective response and evaluation of the stimuli. Here we provide evidence for a causal link between mere exposure to fictitious news reports and subsequent voting behavior. In four pre-registered online experiments, participants browsed through newspaper webpages and were tacitly exposed to names of fictitious politicians. Exposure predicted voting behavior in a subsequent mock election, with a consistent preference for frequent over infrequent names, except when news items were decidedly negative. Follow-up analyses indicated that mere media presence fuels implicit personality theories regarding a candidate's vigor in political contexts. News outlets should therefore be mindful to cover political candidates as evenly as possible.

**Data Availability Statement:** OSF project repository: https://osf.io/ue7fb/ (doi: 10.17605/OSF.IO/UE7F) Pre-registration Experiment 1: https://aspredicted.org/h6b2u.pdf Pre-registration Experiment 2: https://aspredicted.org/8wk74.pdf

## Introduction

Democratic elections require open and accessible media. Impartial media coverage fosters informed voting decisions by allowing politically engaged voters to evaluate a candidate based on relatively objective information. Media coverage, however, may even affect those media consumers that pay only fleeting attention to the content of the reports and cast their vote based on heuristic decision-making [1, 2]. This hypothesis builds on classic observations of the mere exposure effect, a subtle but robust change in the affective evaluation of stimuli that are encountered repeatedly [3–9]. Mere exposure likely derives from increased processing fluency that is primed by repeatedly experiencing an event or object [10–13]. Correlational data from panel surveys suggest that this mechanism may indeed affect voting by showing that media presence predicts election outcomes [14, 15]. Determining whether there is indeed a causal influence of media presence requires experimental methods, however. We therefore aimed at complementing previous survey methodology by an experimental approach to determine whether and how mere exposure to media reports influences political elections.

Pre-registration Experiment 3: https://aspredicted.org/dv74d.pdf Pre-registration Experiment 4: https://aspredicted.org/ks9tv.pdf

**Funding:** The publication was supported by the Open Access Fund of Universität Trier and University of Würzburg and by the German Research Foundation (DFG). R.P. is funded by a Heisenberg grant of the DFG (PF 853/10-1).

**Competing interests:** The authors have declared that no competing interests exist.

## Mere exposure and voting

A large body of psychological research has established that the more often people encounter a particular stimulus, the more positively they come to evaluate it [7, 8]. This mere exposure effect occurs across species and cultures, and it even occurs for stimuli that are presented subliminally (for reviews and meta-analyses, see [5, 8]). Not surprisingly, political campaigns try to leverage this effect by maximizing a candidate's visibility and presence in both online and offline channels.

The effectiveness of such persuasive attempts has been tested, among others, in a recent field experiment via Twitter [16]. In an experimental group, users were asked to follow the tweets of a selected politician alongside the tweets of two additional political Twitter accounts, whereas participants in a control condition were only exposed to tweets of the two additional accounts. After receiving the corresponding Tweets for a month, exposure to the selected politician had indeed increased positive feelings toward the candidate for the experimental group relative to the control group. However, this study did not find an effect on people's voting behavior in the election. These results mirror other recent assessments of the relatively limited effects of direct exposure to political campaigns in online media [17–20], especially when compared to robust effects of mere exposure to traditional means of political campaigning (e.g., [21]).

Exposure to political candidates is not limited to messages that are sent directly by a political actor or the corresponding party, however, as is the case for conventional yard signs, election posters, and online Tweets alike [18, 22]. A second major source relates to media coverage in newspapers, television, radios, and online media [23]. Information from such media outlets has a different quality than information from political campaigns, because it is less likely to be perceived as biased and a direct persuasive attempt. Rather it builds an overarching narrative of current events in which political information becomes embedded [24]. Exposure effects due to media coverage might therefore differ from the impact of direct exposure to political campaigns [25], and reports of public opinion polls have sometimes been observed to act as self-fulfilling prophecies by kindling emerging trends [26]. Even when not discussed in terms of public opinion polls, simply mentioning a certain candidate might still affect voting behavior, and such an effect has indeed been suggested by correlational observations in large-scale panel survey [14]. Exploring this question is especially relevant when considering that many current news outlets focus their reporting on a limited number of political actors. Fig 1 shows an example for the prevalence of high-profile political figures in new reports. This tendency mirrors current trends of content concentration across different types of news outlets [27]. Furthermore, panel data suggests a relation between media visibility and voting behavior, partly mediated by the affective tone of media reports [14]. The present experiments therefore aimed at documenting a causal role of such incidental exposure on voting behavior.

## Current research

In the current work, we implemented an experimental design to model the impact of mere exposure on media consumers who are exposed to reports on political candidates via online news. Each participant browsed through a series of fictitious online newspapers containing a picture, a headline, a corresponding catchphrase, and information on the market trends for five fictitious companies (S1 Fig). Participants were encouraged to monitor these market trends because they could later invest in one of the stocks to obtain additional rewards if choosing the most promising company. The content of the remaining news items was irrelevant for the participants' task so that they did not have to attend to the information provided. Still, each headline featured one of two names (*Smith* vs. *Jones*) with one name appearing frequently and one name appearing infrequently. Upon completing the newspaper task

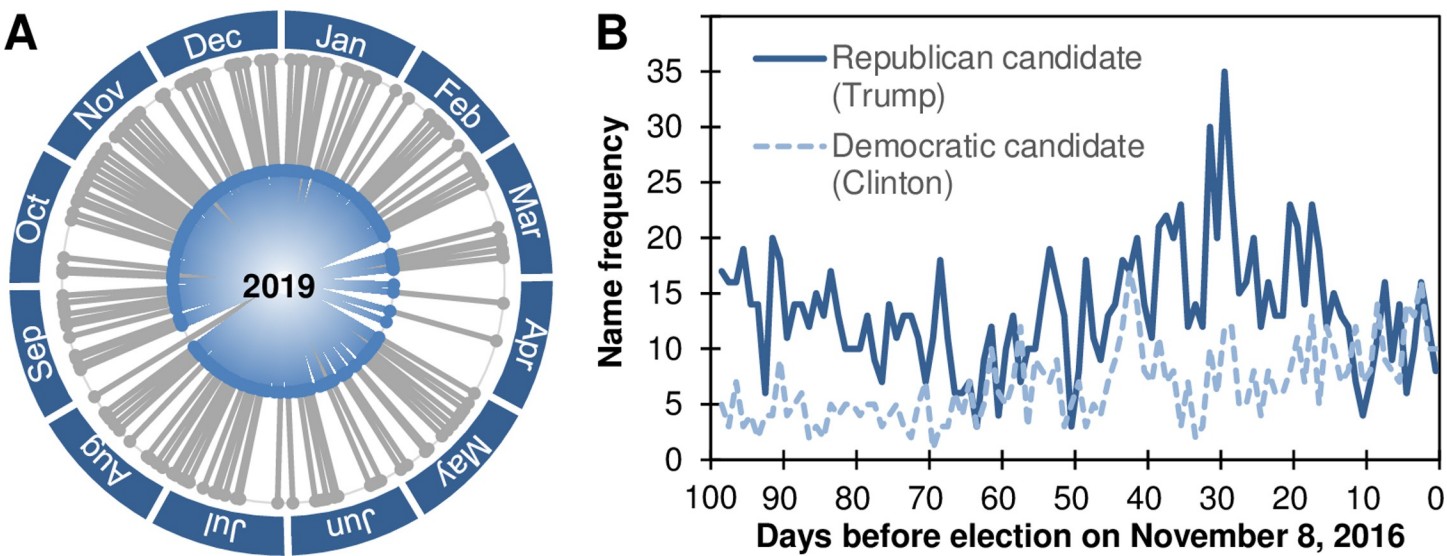

**Fig 1. Exemplary focus of news outlets on single political figures. (A)** Grey lines on the outer circle mark days in which the lead headline of the New York Times webpage featured the name of the president of the United States of America (38%) during 2019. Blue lines in the inner circle mark days in which the name appeared in the corresponding catchphrase (50%). These data showcase a remarkably high media presence for an extended period of time. **(B)** Appearance of the Republican candidate and the Democratic candidate on the homepage of the New York Times before the 2016 presidential election in the United States of America. Numbers on the y axis indicate the absolute frequency across the entire main webpage. **Method:** Data were gathered from the Internet Archive's Wayback Machine (http://www.archive.org/web). For each day of 2019 we assessed the snapshot closest to noon (12:00). No backup was available for two days which were coded as missing. For the remaining days we assessed whether the name of the US president appeared in the headline or catchphrase of the target article. Similar statistics were gathered for the 100 days leading up to the 2016 presidential election (August 1, 2016 –November 8, 2016). Missing data for one day in the timeline were interpolated by computing the average of the preceding and following day.

participants were then asked to imagine that the newspaper's country would hold elections for a political leader and they were asked to cast their vote.

We predicted an above-chance preference for the frequent name as compared to the infrequent name, and we tested this hypothesis in four pre-registered experiments as summarized in Fig 2. To exclude potential confounds relating to the name identity, we randomly allocated the two names to the frequency conditions, randomized the position of both names on the ballot paper across participants, and assessed the participants' strategies in a structured debriefing. To build a convincing database, we decided to conduct a series of experiments that used a highly similar design but focused on varying the valence of the news items. In a nutshell, Experiment 1 used neutral to slightly positive news items and showed a strong preference for the name that appeared frequently in these news. Experiment 2 repeated the same setup with decidedly negative news items and did not find a strong preference for either candidate. Experiment 3 and 4 assessed a setting with only neutral items, again eliciting a strong tendency to vote for the frequent name.

## Methods

### Experimental methodology: Overview

Each experiment featured three stages: Participants commenced with a newspaper task, they then cast their ballot and finished by answering a set of debriefing questions. All participants were recruited via Amazon Mechanical Turk and the experiment was run on a dedicated private web server. Participants received a written consent form prior to starting the experiment. Individual ethics approval was not required for this online study as per the guidelines of the Ethics Committee of the Institute for Psychology of the University of Wuerzburg. All

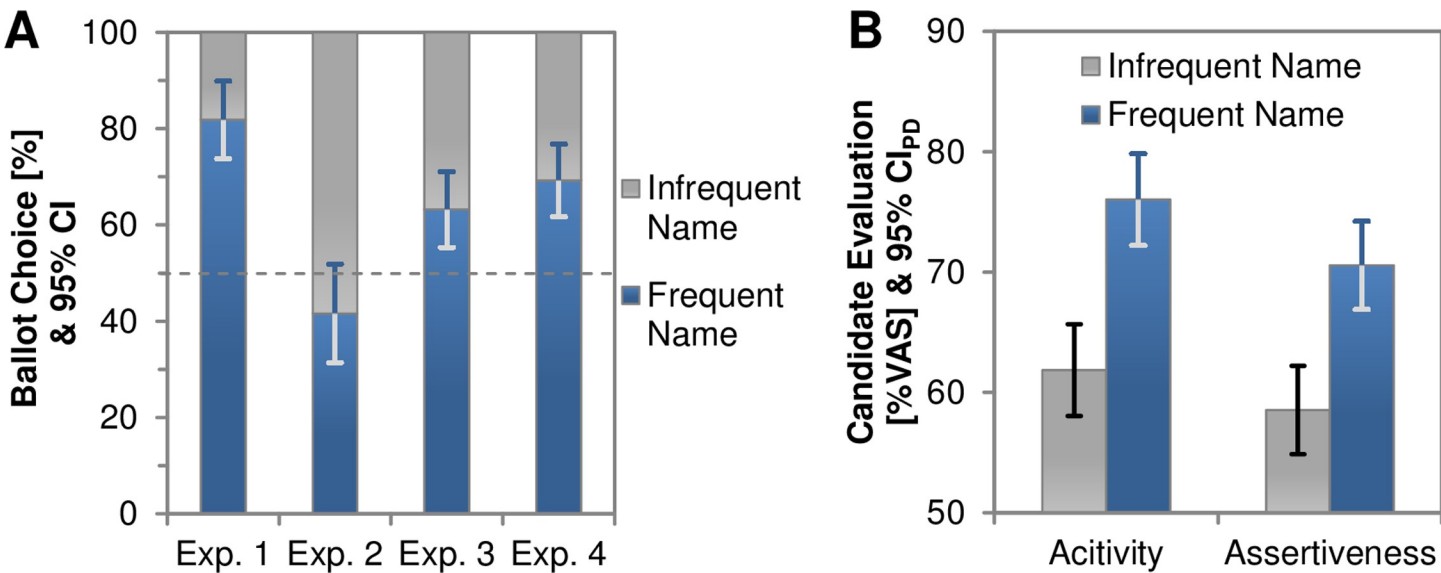

**Fig 2. Mere exposure effects on voting behavior. (A)** Experimental evidence for a direct effect of media exposure on voting behavior. Participants were given the choice between a name that they had frequently encountered when browsing through a series of fictitious news webpages and a name that they had encountered only infrequently. Experiment 1 featured neutral to positive headlines, Experiment 2 employed decidedly negative news coverage, whereas the remaining experiments drew on strictly balanced valences. The frequent name appeared in 95% of the headlines in Experiment 1–3 and it appeared in all headlines of Experiment 4. Error bars show 95% confidence intervals for the proportion of votes for the frequently appearing candidate. **(B)** Evaluation of the two candidates regarding activity and assertiveness on Visual Analogue Scales (VAS) based on the pooled data of Experiment 3 and 4. Error bars show 95% confidence intervals for paired differences [31].

experiments were conducted in accordance with these guidelines, thus also implementing the ethical guidelines of the German Psychological Society and the Declaration of Helsinki.

## Newspaper task

Participants received a short introductory instruction with the following wording:

> "For this experiment, please imagine living in the fictional country called 'Rifaa'. In the following you will see 40 pages of the newspaper 'Rifaa News'.

> On each page you will see the price development of five stocks. Your task is to monitor these developments and at the end you have to invest in one of these five stocks. If you choose the stock that had the best performance, you will earn a bonus of 50 Cents."

S1 Fig shows a sample page. Each page contained the label of the newspaper, RIFAA NEWS, in bold type, followed by a date starting on January 20, 2017, that increased by one day each time participants advanced to the next page. The next section of the page contained a picture, a headline and a corresponding catchphrase. The pictures were taken from the Pixabay library (Pixabay GmbH, Berlin) and complete lists of the stimulus material are available in the OSF project repository linked in the *Open Practices Statement*. The final section of the page showed the names of five fictitious companies–labelled Freerider Holding, Frontpage Business Analytics, Rifaa Logging and Steel, rifAir Transportation, and technoForm. Each name was accompanied by either a green upward arrow signaling positive stock trends or a red downward arrow signaling negative stock trends. One of the companies showed a positive trend in 25% of the trials, three companies showed a positive trend in 50% of the trials, and one company showed a positive trend in 75% of the trials, with the allocation of names to success rates being randomized across participants.

Participants cycled through the series of news pages by clicking a "Next" button in the bottom-right corner of the screen. A progress bar in the lower center of the screen indicated the remaining news pages and participants were given the opportunity to select one of the companies after going through the news pages. They received immediate feedback on the correctness of their investment choice, including the accompanying bonus when investing in the most profitable stocks.

Each headline contained either the name *Smith* or the name *Jones*. One of the names occurred frequently (see the individual experiments for exact frequencies) whereas the other name appeared infrequently, and we randomized the allocation of names to the two frequency conditions across participants. The content of the newspaper was drawn from a pool of forty matching sets of pictures, headlines, and catchphrases (see the OSF project repository) and presented in random order.

Several methodological considerations guided this setup, based on comprehensive accounts of the literature [5, 28, 29]. These considerations resulted in a conservative experimental approach that is maximally likely to generalize to real-world voting behavior. This especially pertains to the number of stimulus presentations, the nature of the stimulus material, and the measurement of mere exposure effects, for which meta-analytic estimates are available [5].

**Number of stimulus presentations.** Previous work on mere exposure has suggested that the effect evolves with unique trajectories for different types of stimuli, ranging from gradual increases in stimulus evaluation to inverted U-shaped functions. Both types of trajectories tend to produce reliable mere exposure effects with about 10 stimulus presentations whereas the effect has been observed to reverse for certain types of stimuli, especially with 30 or more presentations. Because real-world media exposure to political candidates likely exceeds the methodological optimum of 10 stimulus presentations we chose to implement 40 trials to ensure that the present results would not be overwritten by a potential reversal of the effect with increasing exposure. Note that meta-analytic evidence suggests 40 presentations to be the exposure frequency with the smallest mere exposure effects on affective evaluation (see Table 2 in [5]) so that the present design appears to be maximally conservative.

**Stimulus material.** Mere exposure effects differ between simple, isolated stimuli such as individual pictographs and more complex stimuli as in the present case [29]. We thus opted to include unnecessary features such as an irrelevant, salient image to mirror the layout of typical news webpages.

**Measurement.** Mere exposure effects are typically assessed via graded scales because such scales increase the chances of observing reliable but small differences in stimulus evaluation. The present measurement via a ballot paper that included only two options hence constrains the power of the experimental design. We still opted to measure mere exposure effects in terms of this conservative way because elections necessarily force a non-graded, categorical decision.

## Simulated election

Immediately after the newspaper task, participants were instructed to imagine an upcoming election in the newspaper's country: "Imagine the people of Rifaa were to elect a new political leader. Which of the following candidates would you vote for?"

To cast their vote, they could click one of two boxes in a horizontal arrangement. One box featured the name Smith and one box featured the name Jones, while the allocation of the frequent vs. infrequent name to the left vs. right box was randomized across participants.

## Debriefing

The experiments concluded with a structured debriefing. Participants were asked to rate their perception of the news articles, the amount of attention they had paid to content other than the market developments, and potential suspicions about the role of this additional content. The three questions were answered on a Visual Analogue Scale (VAS). Each scale had a short description and verbal anchors at both poles: (a) "The news articles in the upper part of each display were . . ." with anchors "a welcome distraction" and "annoying"; (b) "How much attention did you pay to information unrelated to the companies' outcomes?" with anchors "I tried to ignore all unrelated items" and "I read each of the headlines carefully"; (c) During the experiment, did you feel like the upper part of the page might be of interest later on?" with anchors "No, I did not expect it to be relevant" and "Yes, I suspected there would be something about the upper part". A final, open-ended question targeted the headlines directly: "Did you notice any pattern in the headlines or catchphrases of the articles?", and participants entered their response to this question in a text box.

We then asked participants to report demographic information though answering these questions was optional. This information comprised the participant's age, gender (male vs. female vs. non-binary vs. prefer not to say), education (in years, including university education), and country of origin. Finally, participants received a survey code to claim their compensation and a short summary on the study's objectives.

## Analysis plan

As per our pre-registration, our main analysis was a $X^2$-test with the null hypothesis of an even distribution of votes across the two names (i.e., with expected frequencies $f_{expected} = \frac{N}{2}$). We preferred this approximate statistical procedure over an exact binomial test because it allows for straightforward extensions to between-experiment comparisons (see Experiment 2 onward) though we provide the corresponding exact $p$-values ($p_{exact}$) in all following analyses as well.

In addition to the analysis of the full data set we re-analyzed subsets of the data (a) by including only full data sets with available demographic information ("Full sets only"), (b) by including only those datasets that did not mention the names *Smith* or *Jones* in the open-ended debriefing question on detected patterns in the headlines ("Name not mentioned"; as a proxy to potential suspicions or awareness of the stimulus material), and (c) by including only those datasets that did not comment on valence in the open-ended debriefing question ("Valence not mentioned").

All analyses in the classical framework of null hypothesis significance testing were supplemented by Bayesian proportion tests. These procedures were implemented via the *BayesFactor* package version 0.9.12–4.2 in R. The corresponding null hypothesis assumed a binomial distribution with success probability $p_{success} = 0.5$ whereas the alternative hypothesis was modeled as a logistic function with an r-scale parameter of 0.5 to credit the medium effect underlying our power calculations (see below).

Follow-up tests were performed via separate, univariate logistic regressions on full data sets. These analyses implemented ballot choice as binary criterion and the participants' response to each of the three debriefing questions (in %VAS) as a metric predictor (reported in the Appendix).

## Sample and statistical power

**Experiment 1.** Effect sizes for mere exposure vary considerably across different settings. We thus based our power calculations on a conservative estimate of medium value of Cohen's $w$ = .30. This suggested a sample size of $N$ = 88 participants for a power of $1\text{-}\beta$ = .80 at $\alpha$ = .05 (computed in R via the *pwr.chisq.test* function of the *pwr* package version 1.3–0).

We collected 91 datasets on Amazon Mechanical Turk and had to remove three participants who seemed to have completed the experiment repeatedly as judged by their answers to the open-ended debriefing questions (including the three datasets into the analyses did not affect the pattern of results). An additional four participants did not disclose their demographic information. The remaining participants had a mean age of 34.0 years (SD = 9.08; range: 23–69), 54 were male, 30 female, and they had spent an average of 12.4 years in education (SD = 6.02). Forty-four participants were from the United States of America, 20 from India, and 7 from other countries; the remaining participants did not state their country of origin.

**Experiment 2.** We collected 92 datasets on Amazon Mechanical Turk and had to remove three participants who had already participated in Experiment 1 according to their MTurk ID. Experiment 1 had yielded a large effect of mere exposure to mildly positive headlines. If this impact had been driven entirely by the valence of the headlines, the sample size of Experiment 2 (using negative headlines) provided a power of $1\text{-}\beta$ > .99 to observe a corresponding effect in the opposite direction (assuming w = .59). An additional four participants did not disclose their demographic information. The remaining participants had a mean age of 39.2 years (SD = 11.39; range: 23–69), 51 were male, 34 female, and they had spent an average of 14.3 years in education (SD = 4.49). Seventy-three participants reported to be from the United States of America, 3 from India, 1 from Italy; the remaining participants preferred not to report their country of origin.

**Experiment 3.** We aimed to increase the sample size as compared to the preceding experiments to account for a potentially reduced effect size as the design now used headlines of mixed valence. We thus collected 158 datasets on Amazon Mechanical Turk and had to remove 14 participants who had already participated in Experiment 1 or 2 according to their MTurk ID or were identified as repeatedly taking part in Experiment 3. An additional six participants did not to disclose their demographic information. The remaining participants had a mean age of 33.8 years (SD = 9.20; range: 20–67), 95 were male, 41 female (8 preferred not to state their gender), and they had spent an average of 14.2 years in education (SD = 4.47). Eighty-eight participants reported to be from the United States of America, 21 from India, 11 from other countries, and 25 preferred not to state their country of origin.

**Experiment 4.** Power analyses suggested a sample size of N = 150 for a high-powered replication of Experiment 3 (assuming w = .264 and $1\text{-}\beta$ = .90), and we thus collected 155 datasets on Amazon Mechanical Turk. From these participants we had to remove 12 participants who had already participated in Experiment 1–3 according to their MTurk ID. An additional participant did not to disclose their demographic information. The remaining participants had a mean age of 35.1 years (SD = 10.11; range: 18–70), 91 were male, 50 female, 1 gender-diverse, 1 preferred not to say, and they had spent an average of 13.9 years in education (SD = 4.87). One-hundred and ten participants reported to be from the United States of America, 5 from Brazil, seven from other countries; the remaining participants did not report their country of origin.

## Results

Fig 2 summarizes the results of the four experiments. When using a set of neutral to moderately positive news items (Exp. 1), participants indeed showed a decisive bias towards voting for the candidate that appeared frequently, 81.8% (N = 88), 95% CI = [73.8; 89.9], $X^2(1)$ =

35.64, $p < .001$, w = .636. This pattern occurred also for participants who reported not to have attended the news items at all, and it was stable for a range of statistically conservative follow-up tests as reported in the Supplementary Material (S2 Fig and S1–S4 Tables). The surprisingly strong effect observed in the first experiment may suggest that, in addition to mere exposure, participants' voting preferences might have been boosted by positive evaluative conditioning, a bias towards positive evaluation of neutral stimuli that had co-occurred with positive information. If the observed effects indeed depended on evaluative conditioning, then using negative information should reverse the pattern, possibly even showing a larger reversed effect for negative news items, because negative events are particularly efficient in capturing attention [30]. In Experiment 2, we thus examined boundary conditions for mere exposure effects on voting behavior by exposing participants to a set of decidedly negative news items. Here, the distribution of votes indicated a slight trend towards favoring the infrequent name, i.e., the name that had been coupled with task-irrelevant negative information less often, though the distribution did not differ significantly from chance level, 41.6% ($N = 89$), 95% CI = [31.3%; 51.8%], $X^2(1) = 2.53$, $p = .112$, w = .169.

To determine whether the effect size for the negative headlines of Experiment 2 differed from the effect size for positive headlines as observed in Experiment 1, we reverse-coded the distribution of the former experiment and compared both experiments with a $X^2$-test for independence. This test indeed suggested that the proportion observed in Experiment 1 deviated more strongly from an even distribution than the proportion observed in Experiment 2, $X^2(1) = 10.45$, $p < .001$, w = .243. This was true even though the negative items of Experiment 2 were more extreme than the neutral to positive items of Experiment 1. A sizeable share of the participants also explicitly commented on the negative tone of the newspaper articles (22/89, as compared to 3/88 in Experiment 1). The distribution of valence-related comments differed significantly between experiments, $X^2(1) = 14.86$, $p < .001$, w = .290, suggesting that the negative headlines of Experiment 2 had a weaker impact, even though their valence was more salient than for the neutral to positive headlines of Experiment 1.

To arrive at a maximally unbiased estimate of mere exposure in the absence of evaluative conditioning we employed a pre-rated subset of the previous news items to feature a balanced mix of valences. We thus asked five raters to judge the valence of the 80 newspaper headlines and catchphrases employed in Experiment 1 and 2. Instructions emphasized valence rather than interest in the material or arousal triggered by it. Inter-rater agreement across all items amounted to Krippendorff's α = .78, 95% CI = [.74, .81] as computed via the *krippendorffsalpha* package v1.0 in R. We then assessed the range of the ratings for each item (maximum ratings–minimum rating across the five raters), standardized the range scores and discarded three items for which the corresponding $z$-scores exceeded 2. The remaining items were matched for valence so that the final selection consisted of 40 headlines and corresponding catchphrases with an even distribution of moderately positive and moderately negative items (see the "Ratings" section of the OSF project repository for details).

Experiment 3 employed this balanced stimulus set and again showed a stable preference for voting for the frequently featured candidate, 63.2% ($N = 144$), 95% CI = [55.3%, 71.1%], $X^2(1) = 10.03$, $p = .002$, w = .264. A fourth and final experiment replicated this pattern in a situation in which participants were exclusively confronted with the frequent name so that the infrequent name did not appear until participants saw the ballot paper, 69.2% ($N = 143$), 95% CI = [61.7%, 76.8%], $X^2(1) = 21.15$, $p < .001$, w = .384.

The latter two experiments further sought to address potential corollaries of the observed exposure effects. Participants were thus asked to rate both political candidates for perceived activity and assertiveness which consistently resulted in higher ratings for the frequent name, $p$s $< .001$, $d_z \geq 0.36$ (Fig 2B). To test whether an individual's perception of the candidates on these

variables predicted voting outcomes, we performed logistic regression analyses on the pooled data of Experiment 3 and 4. Separate analyses were run for activity and assertiveness, respectively, and we used the rating difference between the frequent name and the infrequent name as predictor (with a value range from -100 to 100) and the voting outcome as criterion (coded 0 when voting for the infrequent name and 1 when voting for the frequent name). Results suggested that a participant's difference in activity ratings between the frequent and the infrequent name indeed predicted their ballot choice with a more pronounced rating for the frequent name coinciding with higher odds of being voted for, $\beta = .009$, $z = 2.13$, $p = .033$. This corresponds to a change in odds ratio of about 1% per difference of 1 point on the rating scale. By contrast, the difference in assertiveness ratings did not predict voting behavior, $p = .364$.

## Discussion

Our results attest to a remarkable influence of media exposure on voting behavior, while follow-up tests suggested that media presence fosters implicit personality theories of how commanding a political figure is [32, 33]. Notably, the observed effect size exceeds the impact of mere exposure in the context of political campaigns, suggesting that the absence of a clear persuasive attempt may boost the impact of seemingly impartial media coverage [34–36].

Even though such effects come with clear boundary conditions (see the *Limitations* section below), the observed effect size indicates that the impact of mere exposure is substantial even when only affecting a subset of the voting population [19]. They thus lend experimental evidence to a potential mechanism that had been highlighted in previous work using survey methodology [14]. Any report on a political figure could thus affect election outcomes to a degree, and improve that candidate's chances simply by mentioning their name. Media outlets should, therefore, aim not only at providing objective information on potential candidates, but they should also be mindful of the frequency at which they report on different candidates.

Interestingly, mere exposure only predicted voting behavior when content was neutral or positive, but not when it was negative. This observation adds to the available database on context effects and moderators of the mere exposure effect. Among others, similar observations of a strong impact of mere exposure in neutral and positive contexts as compared to reversed but weak effects in negative contexts have been reported for exposure to faces [37] as well as for exposure to aesthetic stimuli such as artwork [38]. It is important to note that this impact is different from exposure to stimuli that are inherently positive or negative in that affectively charged stimuli have been shown to yield clear polarization with increasingly positive evaluation of positive stimuli and increasingly negative evaluation of negative stimuli after repeated exposure (e.g., [39]). Similar effects have been reported from survey data in the context of elections [40]. Investigating such context effects constitutes a promising avenue for future research, especially because many types of exposure in real-world settings will often be embedded in emotional contexts. A similarly relevant avenue for future research would target exposure frequencies that exceed the number of exposures that can be realized in a typical experimental session. In fact, media coverage of high-profile political figures will routinely exceed tens of thousands of exposures, and current quantitative models cannot account for such situations easily [5]. This highlights that the study of mere exposure in selected fields, such as in the context of voting behavior, will not only come with practical insights but may also inform our theoretical understanding of the dynamics of this classic psychological phenomenon.

Available models on the mere exposure effect, as well as the current database (including the present study), apply particularly to situations with relatively unknown political candidates and limited exposure. This includes local government elections, as well as races where

partisanship is entirely unavailable as is the case when electing mayors, county sheriffs, court clerks, public defenders, school board members, city councils, etc. [17]. Similarly, political primaries especially for congressional races tend to include a range of candidates with voters having limited knowledge about each contender. In such election races, partisan effects are irrelevant, and simple name exposure may be critical with exposure even in selected media reports affecting election outcomes. This impact will often derive from recognition heuristics in that voters will prefer names they recognize over unrecognized names [41, 42], but the present results suggest that the frequency of exposure might affect voting behavior even if voters recognize both names of the contenders or none. Because a large number of electoral decisions are made in contexts such as the ones described above, the present research has direct practical implications by experimentally demonstrating that subtle exposure to political candidates increases voting behavior toward them. This effect partly reflected an impact on implicit personality theories about the candidates' activity, and might further derive from relative salience, which has recently been suggested to mediate mere exposure effects in general [43].

### Limitations and future directions

An obvious limitation of the present experiments is that we deliberately focused on a fictitious situation for which none of the participants could have any pre-existing tendencies or affiliations. Our findings thus likely do not apply to partisan elections and partisan voters but mainly to voters who are not set to vote on a specific candidate or party [44]. Moreover, mere exposure can be expected to emerge especially for those people who pay limited attention to the actual content of media reports, thus being maximally biased by fleeting mention of names instead of building an informed opinion about a candidate's profile [45]. Because the extent of partisan voting differs across electoral systems, being particularly high in voting systems with proportional representation [46], the present results will not apply equally strongly to all types of elections. Using the present approach to compare the impact of mere exposure across different types of elections and across different levels of election relevance for the voter thus appears to be a fruitful avenue for future research.

A second characteristic of the present design is its focus on a setting with two opposing candidates with markedly different exposure frequencies, i.e., one name dominating news coverage while the other name only appeared rarely or did not make any appearance at all. Future work would be well advised to explore the possible parameter space for additional variations by including less biased frequencies and/or multiple competitors. News coverage for these competitors may further be directly related to ongoing elections or it might represent election-unrelated news. Both types of news items may well come with distinct effects of mere exposure on voting decisions. It is also an open question whether mere exposure works similarly for parties or party-generated lists of candidates as it does for individual names. It is also plausible that mere exposure effects differ across nations as a function of their electoral system. The present sample sizes and recruitment strategies do not allow for a comprehensive assessment of such potential cross-cultural differences. The experimental setup introduced by the present work, however, provides a flexible blueprint for how to assess such intriguing topics.

### Conclusions

The present set of experiments yielded strong evidence for a direct effect of exposure to candidate names in the media on voting preferences. This effect emerged for positive and neutral reporting alike, suggesting that the effect applies to a wide range of situations. Crucially, these observations extend previous results from survey studies by providing data from controlled experiments, thus attesting a causal link between mere exposure and voting behavior.

## Open practices statement

All measures, manipulations and exclusions in the study are reported. All experiments were pre-registered and we provide full access to the raw data, analysis scripts and code for replicating the experiments on the Open Science Framework (OSF) as per the following links:

- OSF project repository: https://osf.io/ue7fb/ (doi: 10.17605/OSF.IO/UE7F)

- Pre-registration Experiment 1: https://aspredicted.org/h6b2u.pdf

- Pre-registration Experiment 2: https://aspredicted.org/8wk74.pdf

- Pre-registration Experiment 3: https://aspredicted.org/dv74d.pdf

- Pre-registration Experiment 4: https://aspredicted.org/ks9tv.pdf

## Supporting information

**S1 Text. Supplementary analyses.**
(DOCX)

**S1 Fig. Sample page of the newspaper task; a "Next" button appeared right below this display so that participants could browse through the pages in a self-paced manner.** They were instructed to monitor the developments on the stock market because they could invest in one of the five companies at the end of the experiment to secure a bonus. The remaining content of the page was irrelevant for the main task. However, each headline contained either the name Smith or Jones and, across the 40 news pages, one of the names appeared frequently whereas the other name appeared infrequently.
(DOCX)

**S2 Fig. Validation analyses for all four experiments including sample sizes for each subset of the data.**
(DOCX)

**S1 Table. Detailed statistics for the validation analyses of Experiment 1, which had employed neutral to positive headlines.** See the section Analysis plan for details on the chose subsets; $p$ refers to the p-value of a $X^2$ test with the null hypothesis of an even distribution of votes, w is the corresponding effect size. These statistics are accompanied by the $p$-value of an exact binomial test ($p_{exact}$) and the Bayes Factor estimate for a corresponding Bayesian proportion test computed with the alternative hypothesis in the numerator ($BF_{10}$).
(DOCX)

**S2 Table. Detailed statistics for the validation analyses of Experiment 2 (negative headlines).** See the caption of S1 Table for details on the reported statistics.
(DOCX)

**S3 Table. Detailed statistics for the validation analyses of Experiment 3 (mixed headlines).** See the caption of S1 Table for details on the reported statistics.
(DOCX)

**S4 Table. Detailed statistics for the validation analyses of Experiment 4 (mixed headlines and a frequency distribution of 100% vs. 0%).** See the caption of S1 Table for details on the reported statistics.
(DOCX)

## Author Contributions

**Conceptualization:** Roland Pfister, Patricia Holzmann.

**Data curation:** Roland Pfister.

**Formal analysis:** Roland Pfister.

**Investigation:** Moritz Reis.

**Methodology:** Roland Pfister, Moritz Reis.

**Project administration:** Patricia Holzmann, Moritz Reis.

**Software:** Roland Pfister, Moritz Reis.

**Supervision:** Roland Pfister, Wilfried Kunde.

**Validation:** Roland Pfister.

**Visualization:** Roland Pfister.

**Writing – original draft:** Roland Pfister, Katharina A. Schwarz.

**Writing – review & editing:** Roland Pfister, Katharina A. Schwarz, Patricia Holzmann, Moritz Reis, Kumar Yogeeswaran, Wilfried Kunde.

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
