## [Decision Letter · Decision Letter 0]

25 Apr 2023

PONE-D-22-29025Headlines Win Elections: Mere Exposure to News Media Alters Voting BehaviorPLOS ONE

Dear Dr. Pfister,

Thank you for submitting your manuscript to PLOS ONE. I am very sorry about the dely. It was a hard task finding enough experts to review the manuscript. After careful consideration, we feel that it has merit but does not fully meet PLOS ONE’s publication criteria as it currently stands. Therefore, we invite you to submit a revised version of the manuscript that addresses the points raised during the review process. Please note that you are required to make substantial changes. It is not an easy task, and I can't guarantee the manuscript will be accepted in the next round. However, I do believe it has merits, and I hope you will decide to revise and resubmit.  

The reviewers gave some very important insights. Please follow their advice and revise the manuscript accordingly. In specific, please offer a more substantiated and elaborated theoretical framework (and explain why we need to examine negative and positive mentions, why is there a reason to assume that the results will differ, and how your framework and results align with what we know about the well studied phenomenon of "bad stronger then good").

Also, please refer to the boundary conditions and limitations of your research, especially those about external validity.

We look forward to receiving your revised manuscript.

Kind regards,

Guy Hochman, Ph.D.

Academic Editor

PLOS ONE

Journal Requirements:

a) Did participants provide their written or verbal informed consent to participate in this study?

Reviewers' comments:

Reviewer's Responses to Questions

**Comments to the Author**

1. Is the manuscript technically sound, and do the data support the conclusions?

Reviewer #1: Yes

Reviewer #2: Yes

2. Has the statistical analysis been performed appropriately and rigorously? 

Reviewer #1: Yes

Reviewer #2: Yes

3. Have the authors made all data underlying the findings in their manuscript fully available?

Reviewer #1: Yes

Reviewer #2: Yes

4. Is the manuscript presented in an intelligible fashion and written in standard English?

Reviewer #1: Yes

Reviewer #2: Yes

5. Review Comments to the Author

Reviewer #1: I have read “Headlines Win Elections: Mere Exposure to News Media Alters Voting Behavior” which reports four experiments testing the effects of incidental newspaper headline mentions of fake candidates Jones and Smith followed by the outcome variable of dichotomous voting preference calculated with a chi-square test (“supplemented by Bayesian proportion tests”—which I am admittedly unfamiliar with).

I was excited to read this article. The title and abstract make it seem fascinating as a study and tremendously impactful, with serious ramifications for democracy, public opinion, media effects, press bias, campaign strategy, cognitive susceptibility to mass-mediated exposure, persuasion, etc. I like the structured, detailed, basic approach to testing mere exposure to news headlines, and the rigorous testing and analysis of this subject matter amazed this reader. The paper never mentions priming, but that would seem to be the theoretical psychological process at work in prompting people to parrot whatever name they see in a headline as the name they choose on a ballot.

The first study looks at the impact of neutral and positive news items. The second looks at negative news items (which did not differ to a statistically-significant degree; despite participants seeming to indicate that the negative coverage was more memorable). The third and fourth studies looked at mention frequency, finding that people voted for a name they had been exposed to and would not vote for a name they had not been exposed to. The findings indicate that even if media coverage were impartial—providing the slightest balance—giving practically equitable space on news pages to one candidate could have stark impact on election outcomes. The findings are thus severely scary for the reality of political media, which are obviously biased in partisan leanings, suggesting that election outcomes could be decided simply by the slant of media and it could be as simple as which candidate garners the most frequent positive name recognition in headlines, not even as complex as issues or policy stances or ideological purity or legislative records or biography or scandals or electoral experience. The big takeaway, as stated on p. 16, is: “mere exposure only predicted voting behavior when content was neutral or positive, but not when it was negative.” And that is a surprising result. The ramification for campaign strategy thus seems that politicians should do their best to garner positive headlines in friendly outlets and worry less about negative depictions of their opponent(s) and worry less about negative things being said about themselves but instead try to emphasize positive elements that even their detractors may agree with. These findings seem logical, I guess, but I can certainly also think of anecdotal exceptions. And psychologically we all know that bad is stronger than good.

All participants were drawn from mTurk, which is unfortunate. This decision is problematic in the current context because the participants want to be rated favorably in order to maintain their status on the platform and to be paid by the study sponsor. Hypothesis guessing is thus rampant. One could envision, therefore, the participants assuming that they are expected to select a candidate for the fake voting which lines up with their exposure. Mere exposure can thus become tautological in a validity threat. The decision to solicit mTurk participants from all over the world (e.g., about 20% of the participants in Study 1, and about 15% in 3, were from India; and 5 people in Experiment 4 were from Brazil and it is practically guaranteed that those participants do not speak English and would not be voting for candidates named either Jones or Smith [but are indeed probably registered voters—because it’s the law in Brazil that everyone must vote in every election]) also seems unfortunate, casting such a wide net as to make the fake electorate even less realistic as readers of this newspaper voting for either Jones or Smith. Most problematic, of course, is that we don’t know if the mTurkers are registered voters. Ecological validity of a study trying to appraise voter behavior is flawed if the respondents do not engage in voting behavior.

The control and randomization exhibited in the experimentation are pristine. The replication component, though, is questionable, if minor tweaks are made to each presentation of the stimulus and then it’s yet another handful of mTurk respondents participating.

The ecological validity is questionable, which is this reader's biggest issue with the paper. Little in the experiment was realistic. Participants were told to “imagine living in a fictional country called ‘Rifaa’ ” and then they cast fake votes between unreal politicians. The goal of the pursuit is generalizable voting behavior in the real-world (p. 7), yet by its nature participants are providing precisely nothing realistic if from the beginning they are told to imagine living in a fictional country. HOWEVER, if our goal is to test the effects of incidental exposure to media reports, and our population of interest is the electorate who hardly pays attention to political discourse and is horribly misinformed about elections—yet votes and wields the power to control election outcomes—then one could argue that this design is indeed realistic. Moreover, if incidental exposure has this sort of influence then we can only imagine the power of, for example, the New York Times to trumpet one candidate’s name positively over another candidate’s name, and for that headline coverage to prod voters who are favorably inclined toward a candidate to crystalize attitudinally and behaviorally, with the media truly having incredible power to rig elections if they were so desirous. (And indeed we hear reports of social media platforms doing this very type thing with partisan users, not to mention journalists getting caught trying to put their hands on the scales.) However, the mere exposure, as I understand the stimuli, was comprised of 40 exposures. I follow the logic of making it higher than 10, and I see the meta-analysis citations, but if a voter is really that ignorant of a campaign and unfamiliar with candidates then it seems contradictory that they would sustain upwards of 40 exposures to a given candidate—not to mention that it seems implausible that they would spend this much time flipping through the pages of newspapers.

A deflating admission appears on p. 15: “Even though such effects come with clear boundary conditions – they will not affect partisan voters.” Most voters are partisan voters. And we are at an historical apex of partisan tension—with such animus toward the outgroup and blind allegiance of favoritism toward the ingroup. The exclusion of partisan consideration in this series of studies is thus an omission presenting a tragic limitation of the findings. This limitation is magnified when one considers in reality voters may expose themselves to media reports with glowing positive coverage of their co-partisan candidate and rotten negative coverage of their opposing candidate, and that frequency would then drive their behavior through self-selection bias.

The distractor task element of the experiments—having respondents think they are putatively paying attention to stock picking amidst the fake newspaper spread—was admirable. And giving the participants immediate feedback on their investment decisions seemed inventive and comprehensive in the distractor task. Good work. I like it when an experiment tests realism—as much as an artifactual stimulus can be realistic—and indeed a study of voting behavior affected by media exposure seems more realistic if participants are reporting responses based on incidental exposure as opposed to explicit/overt instructions prompted by researchers.

As a sidenote, this study reminded me of research that has shown that public opinion polling results have the power to be self-fulfilling prophecies—with implications for rigging polls to boost or depress election turnout.

The material presented on pp. 24-25 is incredible. I cannot imagine how much time and work must have gone into compiling, calculating, and reporting those data. The presentation is impressive and very well-intentioned. However, what it really boils down to, in the eyes of this reader, is unnecessary and uninterpretable. Notice that Figure A cannot be interpreted as meaning anything, nor can Figure C. They are so small, with the colors all mingling together. I would save space in the manuscript by eliminating that content from the paper. The material is also unnecessary. We don’t need to be told that some candidates get far more media attention than others (in her own book about the 2016 election, “What Happened,” Clinton has no shortage of pages committed to complaining that Trump got more press coverage than her), or that some candidates sustain more negative press coverage than others (and HCR complains in “What Happened” that she seemed to draw more negative coverage for numerous alleged misdeeds, such as deleting 30,000 e-mails of international/national security import from when she was Secretary of State and saved them on a private computer server which may have been hacked by foreign actors). I would perhaps save that content for some other project rather than expending it on this manuscript, and hopefully in future usage more space can be committed to it so it can be larger to be able to read and for readers to be able to make sense of.

One of the citations in the References – Hopmann et al. (2010) https://www.tandfonline.com/doi/abs/10.1080/10584609.2010.516798 which assessed news bulletins and daily survey data from a national election in Denmark -- may have already reported what was found in the present paper: “it is found that the more visible and the more positive the tone toward a given party is, the more voters are inclined to vote for this party.”

Typos:

p. 4 – “Information from such media outlets has a different quality than information from political campaigns, because it is less likely to be perceived as biased and direct persuasive attempt.” – I think you meant to have the word “a” in there: *as a biased*

pp. 10–11 – I don’t think you mean “datasets”; I think you mean *participants* or *respondents*

p. 11 – “An additional six participants did not to disclose”; delete the word ‘to’

p. 12 – “participants who reported not to have attended the news items at all”; missing a word, perhaps ‘to’—thus: “attended to the news items”

In Figure S2 we see an example of the stimulus, a fake news headline, stating “Jones inviting townsmen to get creative” – which is presumably a positive headline for candidate Jones. I wonder how much ecological validity this has. Political candidates do not typically glean media coverage during an election season for touting that painters get creative; the media tend to focus on “horse race” coverage, or on issue rollouts, or other markers of electability; albeit as noble as it seems to aspire for this sort of uplifting campaign coverage in a newspaper.

I had such high hopes for this paper. I really wish the participants had been real voters—not mTurkers from the U.S. and India and Brazil and whereever else—and that the design had been more realistic of actual media exposure to actual politicians in a real election. I cannot praise the authors enough, though, for all the comprehensive and thoughtful and robust work that has gone into this project; amazing work, really.

Reviewer #2: The manuscript “Headlines Win Elections: Mere Exposure to News Media Alters Voting Behavior” reports four preregistered web studies that explore the effect of media exposure on voting behavior. In all studies, participants were cued into a voter perspective and were given the choice in a mock election between a name that they had frequently encountered when browsing through a series of fictitious news webpages and a name that they had only encountered infrequently (or not at all). Experiment 1 presented neutral to positive headlines, Experiment 2 employed decidedly negative news coverage, whereas the remaining experiments (3 and 4) drew on strictly balanced valences. The results indicated that exposure predicted voting behavior in a subsequent mock election, with a consistent preference for frequent over infrequent names, except when news items were decidedly negative (Experiment 2). In addition, the authors found (Experiments 3-4) that the participants’ differences in activity ratings between the frequent and the infrequent name constituted a significant predictor of voting behavior.

I found this topic to be an interesting and valuable line of research with important practical implications. Specifically, the manuscript makes a unique contribution to the state of the art by examining the mere exposure effect with a well-designed (and creative) manipulation and under control settings that simulated voting behavior. In addition, the experiments also contribute to what is known about the boundary condition of this effect, and its potential mechanisms. That said, I think that some points in the theoretical and empirical sections of the manuscript need to be clarified and further developed. I believe the authors should give further attention to the points below to increase the impact of the paper and its accessibility to a wider audience.

1. The introduction is focused and concise, but I think the authors should elaborate further on previous explanations of the mechanisms underlying the mere exposure effect and their relevance to the current study and hypotheses.

2. The authors chose to report on all the four experiments in one section (e.g., one results section for all the experiments). As a reader I found it difficult to understand the differences between the experiments (in fact I only figured it out at the end of the results section and after careful reading of the caption to Figure 1). The authors should thus include an overview of the experiments much earlier, which describes the order of the experiments, and explain more clearly the differences, the specific contribution, and the theoretical rationale for each experiment. .

3. Generally, I think the method used in this study is original and clean. However, one main concern is that the frequent name appeared in 95% of the headlines in Experiment 1-3 but in all headlines in Experiment 4. The 100% vs. 0% exposure in Experiment 4 creates a very extreme situation that clearly differs from the examples in the reality described in Figure s1B on p. 24 (Trump vs. Clinton). Adding an experiment which includes different levels of certainty for the frequent name (e.g., 55%, 65%, 75%, 85%, 95%) would strengthen the internal and external validity of the findings and be more informative in terms of its threshold and boundary conditions.

4. At the end of the discussion, the authors write: “This effect is partly mediated by an impact on implicit personality theories about the candidates’ competence”. However, the results only showed a weak correlation with activity ratings and no correlation with assertiveness ratings. In addition, if the authors want to justify using a mediation model they should subject this model to a specific statistical test for mediation.

5. Finally, do the results of this laboratory study bear out the claim made in the title “Headlines Win Elections”? I am afraid that this conclusion is too strong since this control lab simulation does not capture some of the most essential features of elections. It is very unusual for all the information on one candidate to be natural or positive or that this candidate will appear 100% of the time compared to 0% for his or her opponent. Even in election races with less prominent political candidates such as municipal elections, voters have more information than names such as pictures of the candidate which have been found to be a very strong predictor (e.g., Todorov et al., 2005). Thus, while the authors clearly demonstrated the potential contribution of mere exposure, the conclusion and title should be toned down.

6. Minor comments – Please check the References (e.g., item 21 lists the name of the first author but no coauthors).

7. Please consider inserting more figures and analyses from the SM into the main text. For instance, Figure s1B p. 24, and the main results of the samples after they were selected by specific variables.

Overall, I believe this work will make an important contribution to the literature, but there are still some concerns that should be addressed. I wish the authors good luck in pursuing this interesting line of research.

6. PLOS authors have the option to publish the peer review history of their article (what does this mean?). If published, this will include your full peer review and any attached files.

Reviewer #1: No

Reviewer #2: **Yes: **Shahar Ayal

---

## [Author Response · Author response to Decision Letter 0]

4 Jul 2023

Please find a reader-friendly version of the following points here:

https://www.dropbox.com/s/kyt491wware3fqr/VotingBehavior_R1_Resp2Revs_final.pdf?dl=0

*** Editor’s summary ***

COMMENT: The reviewers gave some very important insights. Please follow their advice and re-vise the manuscript accordingly. In specific, please offer a more substantiated and elaborated theoretical framework (and explain why we need to examine negative and positive mentions, why is there a reason to assume that the results will differ, and how your framework and results align with what we know about the well studied phenomenon of "bad stronger then good").

Also, please refer to the boundary conditions and limitations of your research, especially those about external validity.

RESPONSE: Thank you very much for the positive assessment. This, of course, extends also to the reviewers who have obviously put substantial effort into their thoughtful reports!

We have expanded on the framing of our work, discussing the added value of experimental contri-butions to a field that has mainly drawn on survey methodology so far, thus offering a complemen-tary and fruitful approach.

In the discussion, we have added a new section titled “Limitations and future directions”, in which we clearly state possible limitations concerning external validity.

*** Reviewer #1 ***

COMMENT: I have read “Headlines Win Elections: Mere Exposure to News Media Alters Voting Behavior” which reports four experiments testing the effects of incidental newspaper headline mentions of fake candidates Jones and Smith followed by the outcome variable of dichotomous voting preference calculated with a chi-square test (“supplemented by Bayesian proportion tests”—which I am admittedly unfamiliar with).

I was excited to read this article. The title and abstract make it seem fascinating as a study and tremendously impactful, with serious ramifications for democracy, public opinion, media effects, press bias, campaign strategy, cognitive susceptibility to mass-mediated exposure, persuasion, etc. I like the structured, detailed, basic approach to testing mere exposure to news headlines, and the rigorous testing and analysis of this subject matter amazed this reader. 

RESPONSE: Thank you for your positive and thoughtful comments!

COMMENT: The paper never mentions priming, but that would seem to be the theoretical psycho-logical process at work in prompting people to parrot whatever name they see in a headline as the name they choose on a ballot.

RESPONSE: We agree and now point to priming mechanisms in the revised introduction. This also includes some classical references to this question (e.g., Jacoby und Whitehouse 1989; But-ler & Berry, 2004) that we introduce right from the get-go in the revision.

COMMENT: The first study looks at the impact of neutral and positive news items. The second looks at negative news items (which did not differ to a statistically-significant degree; despite par-ticipants seeming to indicate that the negative coverage was more memorable). The third and fourth studies looked at mention frequency, finding that people voted for a name they had been exposed to and would not vote for a name they had not been exposed to. The findings indicate that even if media coverage were impartial—providing the slightest balance—giving practically equitable space on news pages to one candidate could have stark impact on election outcomes. The findings are thus severely scary for the reality of political media, which are obviously biased in partisan leanings, suggesting that election outcomes could be decided simply by the slant of me-dia and it could be as simple as which candidate garners the most frequent positive name recogni-tion in headlines, not even as complex as issues or policy stances or ideological purity or legisla-tive records or biography or scandals or electoral experience. The big takeaway, as stated on p. 16, is: “mere exposure only predicted voting behavior when content was neutral or positive, but not when it was negative.” And that is a surprising result. The ramification for campaign strategy thus seems that politicians should do their best to garner positive headlines in friendly outlets and worry less about negative depictions of their opponent(s) and worry less about negative things being said about themselves but instead try to emphasize positive elements that even their detractors may agree with. These findings seem logical, I guess, but I can certainly also think of anecdotal excep-tions. And psychologically we all know that bad is stronger than good.

RESPONSE: We appreciate the thoughtful engagement with the work. It is certainly true that bad/negative material tends to capture attention much more strongly than good/positive infor-mation. This is why we felt it would be important to study positive as well as negative news cover-age. If assuming that only valence affects voting behavior, possibly in the suspected “bad is stronger than good” manner, we would therefore expect positive headlines to have a slight positive effect on election outcomes whereas negative headlines should have a strong negative effect. However, this is not what happened: Positive headlines had a robust positive effect whereas nega-tive headlines did not exert a strong effect at all! This indicates that mere exposure is indeed at work here. We now discuss this point more explicitly when comparing the results of Experiment 1 and 2, also referring to classic work in this field (i.e., Baumeister et al.’s “Bad Is Stronger Than Good”).

COMMENT: All participants were drawn from mTurk, which is unfortunate. This decision is prob-lematic in the current context because the participants want to be rated favorably in order to main-tain their status on the platform and to be paid by the study sponsor. Hypothesis guessing is thus rampant. One could envision, therefore, the participants assuming that they are expected to select a candidate for the fake voting which lines up with their exposure. Mere exposure can thus be-come tautological in a validity threat. The decision to solicit mTurk participants from all over the world (e.g., about 20% of the participants in Study 1, and about 15% in 3, were from India; and 5 people in Experiment 4 were from Brazil and it is practically guaranteed that those participants do not speak English and would not be voting for candidates named either Jones or Smith [but are indeed probably registered voters—because it’s the law in Brazil that everyone must vote in every election]) also seems unfortunate, casting such a wide net as to make the fake electorate even less realistic as readers of this newspaper voting for either Jones or Smith. Most problematic, of course, is that we don’t know if the mTurkers are registered voters. Ecological validity of a study trying to appraise voter behavior is flawed if the respondents do not engage in voting behavior.

RESPONSE: We have added a new section called “Limitations and future directions” in which we discuss potential sampling biases. We would consider the diversity of the sample in terms of na-tionalities a strength rather than a weakness, however. Indian participants, for instance, may not vote in the USA, but they may well vote in India. English also is an official language in India, and in several states, it is used alongside the native language of the state in elections among other gov-ernmental operations. Voting systems also differ across countries (or states within a country) in whether and how voters have to register. Nevertheless, even though we obviously cannot assess these differences empirically in the current setup, sampling people from diverse backgrounds ap-pears to be a reasonable first step to arrive at general rules of thumb. We fully agree, however, that it is a fruitful direction to address such differences empirically, and we point to this opportunity in the discussion.

Finally, to alleviate potential concerns specifically regarding participants from India and Brazil, we re-ran the main analysis for all four experiments with two additional participant selections. First, we removed participants from these two countries specifically. Second, we focused on participants who had stated their country as USA or United States of America (discounting participants who stated, e.g., Maryland or Florida as their country). The results were remarkably robust against these selections with Experiment 1, 3, and 4 always returning significant results and Experiment 2 yielding a non-significant trend as in the analyses reported in the paper. 

 Selection 1: w/o Indians and Brazili-ans Selection 2: Only US-Americans

Experiment---χ²(1)---p---χ²(1)---p

1---21.60---.001---17.79---.001

2---2.85---.091---2.80---.094

3---13.23---.001---16.20---.001

4---17.19---.001---8.65---.003

COMMENT: The control and randomization exhibited in the experimentation are pristine. The repli-cation component, though, is questionable, if minor tweaks are made to each presentation of the stimulus and then it’s yet another handful of mTurk respondents participating.

RESPONSE: The idea of these replications was indeed to be as close to the first study design to establish whether the pattern of results is robust. In our view, this needs to be established before committing to more substantial variations of an experimental paradigm. However, we see that we did not explain this reasoning in sufficient detail in the original version and have justified this meth-odological choice in the revision.

COMMENT: The ecological validity is questionable, which is this reader's biggest issue with the paper. Little in the experiment was realistic. Participants were told to “imagine living in a fictional country called ‘Rifaa’ ” and then they cast fake votes between unreal politicians. The goal of the pursuit is generalizable voting behavior in the real-world (p. 7), yet by its nature participants are providing precisely nothing realistic if from the beginning they are told to imagine living in a fictional country. HOWEVER, if our goal is to test the effects of incidental exposure to media reports, and our population of interest is the electorate who hardly pays attention to political discourse and is horribly misinformed about elections—yet votes and wields the power to control election out-comes—then one could argue that this design is indeed realistic. Moreover, if incidental exposure has this sort of influence then we can only imagine the power of, for example, the New York Times to trumpet one candidate’s name positively over another candidate’s name, and for that headline coverage to prod voters who are favorably inclined toward a candidate to crystalize attitudinally and behaviorally, with the media truly having incredible power to rig elections if they were so de-sirous. (And indeed we hear reports of social media platforms doing this very type thing with parti-san users, not to mention journalists getting caught trying to put their hands on the scales.) How-ever, the mere exposure, as I understand the stimuli, was comprised of 40 exposures. I follow the logic of making it higher than 10, and I see the meta-analysis citations, but if a voter is really that ignorant of a campaign and unfamiliar with candidates then it seems contradictory that they would sustain upwards of 40 exposures to a given candidate—not to mention that it seems implausible that they would spend this much time flipping through the pages of newspapers.

A deflating admission appears on p. 15: “Even though such effects come with clear boundary con-ditions – they will not affect partisan voters.” Most voters are partisan voters. And we are at an historical apex of partisan tension—with such animus toward the outgroup and blind allegiance of favoritism toward the ingroup. The exclusion of partisan consideration in this series of studies is thus an omission presenting a tragic limitation of the findings. This limitation is magnified when one considers in reality voters may expose themselves to media reports with glowing positive coverage of their co-partisan candidate and rotten negative coverage of their opposing candidate, and that frequency would then drive their behavior through self-selection bias.

RESPONSE: We appreciate the feedback and we now cover these points in the new section on “Limitations and future directions”:

“An obvious limitation of the present experiments is that we deliberately focused on a fictitious situation for which none of the participants could have any pre-existing tendencies or affiliations. Our findings thus likely do not apply to partisan elections and partisan voters but mainly to voters who are not set to vote on a specific candi-date or party [44]. Moreover, mere exposure can be expected to emerge especially for those people who pay limited attention to the actual content of media reports, thus being maximally biased by fleeting mention of names instead of building an informed opinion about a candidate’s profile [45]. Because the extent of partisan voting differs across electoral systems, being particularly high in voting systems with proportional representation [46], the present results will not apply equally strongly to all types of elections. Using the present approach to compare the impact of mere exposure across different types of elections and across different levels of election relevance for the voter thus appears to be a fruitful avenue for future research.

A second characteristic of the present design is its focus on a setting with two oppos-ing candidates with markedly different exposure frequencies, i.e., one name dominat-ing news coverage while the other name only appeared rarely or did not make any appearance at all. Future work would be well advised to explore the possible parame-ter space for additional variations by including less biased frequencies and/or multiple competitors. News coverage for these competitors may further be directly related to ongoing elections or it might represent election-unrelated news. Both types of news items may well come with distinct effects of mere exposure on voting decisions. It is also an open question whether mere exposure works similarly for parties or party-generated lists of candidates as it does for individual names. It is also plausible that mere exposure effects differ across nations as a function of their electoral system. The present sample sizes and recruitment strategies do not allow for a comprehen-sive assessment of such potential cross-cultural differences. The experimental setup introduced by the present work, however, provides a flexible blueprint for how to as-sess such intriguing topics.”

While we agree that partisanship could indeed be very important and is not considered in this work, it is worth noting that partisanship effects do not emerge in many real-world elections even within the USA. For example, the elections of county sheriffs, court clerks, public defenders, and mayors tend to involve no political party. Similarly, political primaries and caucuses in the US in-volve candidates from the same political party competing to get the party nomination removing partisanship effects and whether a candidate wins these races or not can be influenced by how much exposure they receive in media. In addition to these elections being devoid of partisan ef-fects, there is also not the kind of high level of name recognition seen in larger election races as people don’t actively follow such races despite millions of dollars being spent on such election races. Our studies map nicely onto such elections and reveal how mere exposure can help candi-dates in such races win elections. We have now added this context into the paper to situate our findings.

COMMENT: The distractor task element of the experiments—having respondents think they are putatively paying attention to stock picking amidst the fake newspaper spread—was admirable. And giving the participants immediate feedback on their investment decisions seemed inventive and comprehensive in the distractor task. Good work. I like it when an experiment tests realism—as much as an artifactual stimulus can be realistic—and indeed a study of voting behavior affected by media exposure seems more realistic if participants are reporting responses based on inci-dental exposure as opposed to explicit/overt instructions prompted by researchers.

As a sidenote, this study reminded me of research that has shown that public opinion polling re-sults have the power to be self-fulfilling prophecies—with implications for rigging polls to boost or depress election turnout.

RESPONSE: Thank you for this thought! We now refer explicitly to this observation when introduc-ing possible effects of media coverage (as compared to direct campaigns).

COMMENT: The material presented on pp. 24-25 is incredible. I cannot imagine how much time and work must have gone into compiling, calculating, and reporting those data. The presentation is impressive and very well-intentioned. However, what it really boils down to, in the eyes of this reader, is unnecessary and uninterpretable. Notice that Figure A cannot be interpreted as meaning anything, nor can Figure C. They are so small, with the colors all mingling together. I would save space in the manuscript by eliminating that content from the paper. The material is also unneces-sary. We don’t need to be told that some candidates get far more media attention than others (in her own book about the 2016 election, “What Happened,” Clinton has no shortage of pages com-mitted to complaining that Trump got more press coverage than her), or that some candidates sustain more negative press coverage than others (and HCR complains in “What Happened” that she seemed to draw more negative coverage for numerous alleged misdeeds, such as deleting 30,000 e-mails of international/national security import from when she was Secretary of State and saved them on a private computer server which may have been hacked by foreign actors). I would perhaps save that content for some other project rather than expending it on this manuscript, and hopefully in future usage more space can be committed to it so it can be larger to be able to read and for readers to be able to make sense of.

RESPONSE: We agree that this material is different from the experimental findings in the main text of the original submission in that it aims at providing graspable everyday observations that motivated our hypothesis in the first place. The supplementary material might therefore not have been the optimal place for reporting the data. At the same time, Reviewer 2 argued in the opposite direction (citing from this review): “„ Please consider inserting more figures and analyses from the SM into the main text. For instance, Figure s1B p. 24, and the main results of the samples after they were selected by specific variables.” In light of the diverging recommendations, we aimed for an intermediate solution: We indeed removed some of the supplementary data entirely from the manuscript as suggested by you (Figure S1C), so that the remaining data is more condensed and more easily accessible. This also allowed us to remove the remaining plots (Fig. S1A-B) to the main text in order to motivate our experiments.

COMMENT: One of the citations in the References – Hopmann et al. (2010) https://www.tandfonline.com/doi/abs/10.1080/10584609.2010.516798 which assessed news bulle-tins and daily survey data from a national election in Denmark -- may have already reported what was found in the present paper: “it is found that the more visible and the more positive the tone toward a given party is, the more voters are inclined to vote for this party.”

RESPONSE: The idea of this study is very similar indeed. The major difference, however, is that Homann et al. used survey data, i.e., correlational observations. We see our study as contributing complementary evidence from an experimental paradigm. The strength of this approach is its abil-ity to document causal mechanisms, which we believe to be a notable contribution to this field. This, of course, comes at the expense of other factors such as limited external validity so that firm conclusions should ideally take both types of studies into consideration. We now make this point clear from the opening paragraph onward. 

COMMENT: Typos:

p. 4 – “Information from such media outlets has a different quality than information from political campaigns, because it is less likely to be perceived as biased and direct persuasive attempt.” – I think you meant to have the word “a” in there: *as a biased*

pp. 10–11 – I don’t think you mean “datasets”; I think you mean *participants* or *respondents*

p. 11 – “An additional six participants did not to disclose”; delete the word ‘to’

p. 12 – “participants who reported not to have attended the news items at all”; missing a word, perhaps ‘to’—thus: “attended to the news items”

RESPONSE: Fixed, thank you!

COMMENT: In Figure S2 we see an example of the stimulus, a fake news headline, stating “Jones inviting townsmen to get creative” – which is presumably a positive headline for candidate Jones. I wonder how much ecological validity this has. Political candidates do not typically glean media coverage during an election season for touting that painters get creative; the media tend to focus on “horse race” coverage, or on issue rollouts, or other markers of electability; albeit as noble as it seems to aspire for this sort of uplifting campaign coverage in a newspaper.

RESPONSE: We now discuss potential implications of election-related vs. election-unrelated cov-erage in the “Limitations and future directions” section (please see the copied paragraph above).

COMMENT: I had such high hopes for this paper. I really wish the participants had been real vot-ers—not mTurkers from the U.S. and India and Brazil and whereever else—and that the design had been more realistic of actual media exposure to actual politicians in a real election. I cannot praise the authors enough, though, for all the comprehensive and thoughtful and robust work that has gone into this project; amazing work, really.

RESPONSE: Thank you very much again!

*** Reviewer #2 ***

COMMENT: The manuscript “Headlines Win Elections: Mere Exposure to News Media Alters Vot-ing Behavior” reports four preregistered web studies that explore the effect of media exposure on voting behavior. In all studies, participants were cued into a voter perspective and were given the choice in a mock election between a name that they had frequently encountered when browsing through a series of fictitious news webpages and a name that they had only encountered infre-quently (or not at all). Experiment 1 presented neutral to positive headlines, Experiment 2 em-ployed decidedly negative news coverage, whereas the remaining experiments (3 and 4) drew on strictly balanced valences. The results indicated that exposure predicted voting behavior in a sub-sequent mock election, with a consistent preference for frequent over infrequent names, except when news items were decidedly negative (Experiment 2). In addition, the authors found (Experi-ments 3-4) that the participants’ differences in activity ratings between the frequent and the infre-quent name constituted a significant predictor of voting behavior.

I found this topic to be an interesting and valuable line of research with important practical implica-tions. Specifically, the manuscript makes a unique contribution to the state of the art by examining the mere exposure effect with a well-designed (and creative) manipulation and under control set-tings that simulated voting behavior. In addition, the experiments also contribute to what is known about the boundary condition of this effect, and its potential mechanisms. That said, I think that some points in the theoretical and empirical sections of the manuscript need to be clarified and further developed. I believe the authors should give further attention to the points below to in-crease the impact of the paper and its accessibility to a wider audience.

RESPONSE: Thank you!

COMMENT: 1. The introduction is focused and concise, but I think the authors should elaborate further on previous explanations of the mechanisms underlying the mere exposure effect and their relevance to the current study and hypotheses.

RESPONSE: We now briefly highlight the contemporary idea of mere exposure deriving from pro-cessing fluency. At the same time, we were reluctant to go into more depth on previous explana-tions such activation or arousal theories that were en vogue in the 1970s (Berlyne, Crandall). We also did not capture later models that assumed the affective response to an object to depend on how many associations this object retrieves from memory. In view of these models, repeatedly encountering an object creates more and more associations that can be retrieved from memory, thus rendering them more affectively charged even if there were only slightly affective responses in each individual encounter. Alternatively, mere exposure has also been explained as a reduction of uncertainty in the sense that unknown objects evoke diffuse and possibly competing perceptual interpretations or evoked action tendencies in the observer (Grush, 1979, mirroring Berlyne’s idea of conflict as uncertainty). Repeatedly encountering the same object paves the way for reducing this conflict by associating the object with a canonical interpretation or action tendency, this ren-dering it more positive.

Even though a more detailed take on such models is interesting on its own right (as is a discussion of the evidence that challenged most models except for the fluency model), we felt that it was too distracting in this case. The current study does not seek to elucidate the mechanisms behind mere exposure (and thus cannot speak to the above issues), but rather it asks whether mere exposure affects voting behavior. If the reviewer does not share this reasoning then we would of course be willing to work the above points into the manuscript, however. 

COMMENT: 2. The authors chose to report on all the four experiments in one section (e.g., one results section for all the experiments). As a reader I found it difficult to understand the differences between the experiments (in fact I only figured it out at the end of the results section and after careful reading of the caption to Figure 1). The authors should thus include an overview of the experiments much earlier, which describes the order of the experiments, and explain more clearly the differences, the specific contribution, and the theoretical rationale for each experiment.

RESPONSE: Apologies for the confusion. We have added an overview paragraph of the four ex-periments and how they connect:

“We predicted an above-chance preference for the frequent name as compared to the infrequent name, and we tested this hypothesis in four pre-registered experiments as summarized in Figure 2. To exclude potential confounds relating to the name identity, we randomly allocated the two names to the frequency conditions, randomized the position of both names on the ballot paper across participants, and assessed the par-ticipants’ strategies in a structured debriefing. To build a convincing database, we de-cided to conduct a series of experiments that used a highly similar design but fo-cused on varying the valence of the news items. In a nutshell, Experiment 1 used neutral to slightly positive news items and showed a strong preference for the name that appeared frequently in these news. Experiment 2 repeated the same setup with decidedly negative news items and did not find a strong preference for either candi-date. Experiment 3 and 4 assessed a setting with only neutral items, again eliciting a strong tendency to vote for the frequent name.”

COMMENT: 3. Generally, I think the method used in this study is original and clean. However, one main concern is that the frequent name appeared in 95% of the headlines in Experiment 1-3 but in all headlines in Experiment 4. The 100% vs. 0% exposure in Experiment 4 creates a very extreme situation that clearly differs from the examples in the reality described in Figure s1B on p. 24 (Trump vs. Clinton). Adding an experiment which includes different levels of certainty for the fre-quent name (e.g., 55%, 65%, 75%, 85%, 95%) would strengthen the internal and external validity of the findings and be more informative in terms of its threshold and boundary conditions.

RESPONSE: We agree that the chosen setting only captures a subset of the many different ways of parametrizing the distribution of frequent and infrequent names. The sketched experiment with a graded manipulation of frequency is highly elegant indeed, but at the same time requires consider-ably large sample sizes. At the same group size per design cell as the in present experiments we’d be looking at around 500-800 participants. This substantial investment would feel like a new project. We therefore decided to discuss this valid point in a new section labeled “Limitations and future directions” where we sketch several promising lines of research that could take the present design to answer additional intriguing questions. 

COMMENT: 4. At the end of the discussion, the authors write: “This effect is partly mediated by an impact on implicit personality theories about the candidates’ competence”. However, the results only showed a weak correlation with activity ratings and no correlation with assertiveness ratings. In addition, if the authors want to justify using a mediation model they should subject this model to a specific statistical test for mediation.

RESPONSE: We have removed the term mediation and replaced “competence” with “activity” to better reflect our analysis here. It now reads: “This effect partly reflected an impact on implicit per-sonality theories about the candidates’ activity...”

Upon revisiting this part of the results section, we also realized that we might not have described the analyses in sufficient detail. We are now more specific in the revision, providing exact coeffi-cient coding and stating that the odds ratio of voting for the frequent name changes at about 1% for each point of the difference score for activity ratings. This effect is not monumentally big, but with a value range from -100 (when rating activity = 0 for the frequent name and activity = 100 for the infrequent name) to 100, we believe that this effect is sufficiently strong to deserve a mention:

“[…] To test whether an individual’s perception of the candidates on these variables predicted voting outcomes, we performed logistic regression analyses on the pooled data of Experiment 3 and 4. Separate analyses were run for activity and assertive-ness, respectively, and used the rating difference between the frequent name and the infrequent name as predictor (with a value range from -100 to 100) and the voting outcome as criterion (coded 0 when voting for the infrequent name and 1 when voting for the frequent name). Results suggested that a participant’s difference in activity ratings between the frequent and the infrequent name indeed predicted their ballot choice with a more pronounced rating for the frequent name coinciding with higher odds of being voted for, β = .009, z = 2.13, p = .033. This corresponds to a change in odds ratio of about 1% per difference of 1 point on the rating scale. By contrast, the difference in assertiveness ratings did not predict voting behavior, p = .364.”

COMMENT: 5. Finally, do the results of this laboratory study bear out the claim made in the title “Headlines Win Elections”? I am afraid that this conclusion is too strong since this control lab simulation does not capture some of the most essential features of elections. It is very unusual for all the information on one candidate to be natural or positive or that this candidate will appear 100% of the time compared to 0% for his or her opponent. Even in election races with less promi-nent political candidates such as municipal elections, voters have more information than names such as pictures of the candidate which have been found to be a very strong predictor (e.g., Todo-rov et al., 2005). Thus, while the authors clearly demonstrated the potential contribution of mere exposure, the conclusion and title should be toned down.

RESPONSE: We are now more explicit about potential limitations throughout and have slightly adapted the title to show that the data is based on fictitious news articles. It now reads: “Headlines win elections: Mere exposure to fictitious news media alters voting behavior”. 

Here is the full, added paragraph on limitations of the study:

“Limitations and future directions

An obvious limitation of the present experiments is that we deliberately focused on a fictitious situation for which none of the participants could have any pre-existing tendencies or affiliations. Our findings thus likely do not apply to partisan elections and partisan voters but mainly to voters who are not set to vote on a specific candi-date or party [44]. Moreover, mere exposure can be expected to emerge especially for those people who pay limited attention to the actual content of media reports, thus being maximally biased by fleeting mention of names instead of building an informed opinion about a candidate’s profile [45]. Because the extent of partisan voting differs across electoral systems, being particularly high in voting systems with proportional representation [46], the present results will not apply equally strongly to all types of elections. Using the present approach to compare the impact of mere exposure across different types of elections and across different levels of election relevance for the voter thus appears to be a fruitful avenue for future research.

A second characteristic of the present design is its focus on a setting with two oppos-ing candidates with markedly different exposure frequencies, i.e., one name dominat-ing news coverage while the other name only appeared rarely or did not make any appearance at all. Future work would be well advised to explore the possible parame-ter space for additional variations by including less biased frequencies and/or multiple competitors. News coverage for these competitors may further be directly related to ongoing elections or it might represent election-unrelated news. Both types of news items may well come with distinct effects of mere exposure on voting decisions. It is also an open question whether mere exposure works similarly for parties or party-generated lists of candidates as it does for individual names. It is also plausible that mere exposure effects differ across nations as a function of their electoral system. The present sample sizes and recruitment strategies do not allow for a comprehen-sive assessment of such potential cross-cultural differences. The experimental setup introduced by the present work, however, provides a flexible blueprint for how to as-sess such intriguing topics.”

COMMENT: 6. Minor comments – Please check the References (e.g., item 21 lists the name of the first author but no coauthors).

RESPONSE: Done.

COMMENT: 7. Please consider inserting more figures and analyses from the SM into the main text. For instance, Figure s1B p. 24, and the main results of the samples after they were selected by specific variables.

RESPONSE: We have moved Figure S1A and S1B to the main text as suggested (as a new Fig-ure 1). We also considered moving Figure S3 (showing the different sub-samples) to the main text but felt that this figure would duplicate too much information from the main results figure (Figure 1 of the original version, Figure 2 in the revision). Also, we felt as if the pattern was robust enough to be discussed in a compact manner, as none of the control analyses pointed towards a meaningful reduction of the observed mere exposure effects.

COMMENT: Overall, I believe this work will make an important contribution to the literature, but there are still some concerns that should be addressed. I wish the authors good luck in pursuing this interesting line of research.

RESPONSE: Thank you!

---

## [Decision Letter · Decision Letter 1]

18 Jul 2023

Headlines win elections: Mere exposure to fictitious news media alters voting behavior

PONE-D-22-29025R1

Dear Dr. Pfister,

We’re pleased to inform you that your manuscript has been judged scientifically suitable for publication and will be formally accepted for publication once it meets all outstanding technical requirements.

Kind regards,

Guy Hochman, Ph.D.

Academic Editor

PLOS ONE

Additional Editor Comments (optional):

Reviewers' comments:

Reviewer's Responses to Questions

**Comments to the Author**

1. If the authors have adequately addressed your comments raised in a previous round of review and you feel that this manuscript is now acceptable for publication, you may indicate that here to bypass the “Comments to the Author” section, enter your conflict of interest statement in the “Confidential to Editor” section, and submit your "Accept" recommendation.

Reviewer #1: All comments have been addressed

Reviewer #2: (No Response)

2. Is the manuscript technically sound, and do the data support the conclusions?

Reviewer #1: Yes

Reviewer #2: Yes

3. Has the statistical analysis been performed appropriately and rigorously? 

Reviewer #1: Yes

Reviewer #2: Yes

4. Have the authors made all data underlying the findings in their manuscript fully available?

Reviewer #1: Yes

Reviewer #2: Yes

5. Is the manuscript presented in an intelligible fashion and written in standard English?

Reviewer #1: Yes

Reviewer #2: Yes

6. Review Comments to the Author

Reviewer #1: I have read through the authors' responses to the reviewers' comments. I am satisfied. I suggest the manuscript proceeds. Thank you

Reviewer #2: I served as Reviewer 2 in the original submission and have now read its revised version. The authors have done a very good job on the revision, and all of my comments have been addressed satisfactorily.

7. PLOS authors have the option to publish the peer review history of their article (what does this mean?). If published, this will include your full peer review and any attached files.

Reviewer #1: No

Reviewer #2: **Yes: **Dr. Shahar Ayal

---

## [Editor Report · Acceptance letter]

21 Jul 2023

PONE-D-22-29025R1 

Headlines win elections: Mere exposure to fictitious news media alters voting behavior 

Dear Dr. Pfister:

I'm pleased to inform you that your manuscript has been deemed suitable for publication in PLOS ONE. Congratulations! Your manuscript is now with our production department. 

Kind regards, 

on behalf of

Dr. Guy Hochman 

Academic Editor

PLOS ONE